# TEST-TIME TRAINING FOR GENERALIZATION UNDER DISTRIBUTION SHIFTS

## ABSTRACT

We introduce a general approach, called test-time training, for improving the performance of predictive models when training and test data come from different distributions. Test-time training turns a single unlabeled test instance into a self-supervised learning problem, on which we update the model parameters before making a prediction on this instance. We show that this simple idea leads to surprising improvements on diverse image classification benchmarks aimed at evaluating robustness to distribution shifts. Theoretical investigations on a convex model reveal helpful intuitions for when we can expect our approach to help.

## 1 INTRODUCTION

Supervised learning remains notoriously weak at generalization under distribution shifts. Unless training and test data are drawn from the same distribution, even seemingly minor differences turn out to defeat state-of-the-art models. Transfer learning, domain adaptation and adversarial robustness are but a few existing paradigms that anticipate differences of sorts between the training and test distribution. In this work, we explore a new take on generalization, without any mathematical structure or data available at training-time to anticipate the distribution shifts.

We start from a simple observation. The unlabeled test instance $x$ presented at test-time gives us a hint about the distribution from which it was drawn. Our approach, called test-time training, allows the model parameters $\theta$ to depend on the test instance $x$, but not its unknown label $y$. This variable decision boundary $\theta(x)$ is powerful in theory (see Appendix A), and raises new challenges in practice that we only begin to address here.

Our algorithm creates a self-supervised learning problem based on this single test instance $x$, updating $\theta$ at test-time before making a prediction. Self-supervised learning uses an auxiliary task that automatically creates labels from unlabeled inputs. For the visual data we work with, the task rotates an image $x$ by a multiple of 90 degrees, and assigns the angle as the label (Gidaris et al., 2018).

Our approach can also be easily modified to work outside the standard supervised learning setting. If several test instances arrive in a batch, we can use the entire batch for test-time training. If test instances arrive online sequentially, we obtain further improvements by keeping the state of the parameters. After all, prediction is rarely a single event. The online setting can be the natural mode of deployment in practice, and shows the strongest improvements.

We experiment with test-time training for generalization under distribution shifts in the context of object recognition on several benchmarks. These include images with diverse types of corruption at various levels (Hendrycks & Dietterich, 2019), video frames of moving objects (Shankar et al., 2019), and a new test set of unknown shifts collected by Recht et al. (2018). Our algorithm does not hurt on the original distribution, but makes substantial improvements under distribution shifts.

In all our experiments, we compare with a strong baseline (labeled joint training) that uses both supervised and self-supervised learning at training-time, but keeps the model fixed at test-time. Very recent work shows that additional *training-time* self-supervision improves robustness Hendrycks et al. (2019b). The joint training baseline we use corresponds to an improved implementation of their work. We are also inspired by Sun et al. (2019), which performs joint training with a self-supervised task in the context of unsupervised domain adaptation, simultaneously on the source and target domain. A comprehensive review of related work follows in section 5.

We complement the empirical results with theoretical investigations in section 4, of when test-time training is expected to help on a convex model, and establish an intuitive sufficient condition, which, roughly speaking, is to have correlated gradients between the loss functions of the two tasks.

## 2 METHOD

Next we describe the algorithmic details of our method. To setup notations, consider a standard $K$-layer neural network with parameters $\theta_k$ for layer $k$. The stacked parameter vector $\boldsymbol{\theta} = (\theta_1, \ldots, \theta_K)$ specifies the entire model for a classification task with loss function $l_m(x, y; \boldsymbol{\theta})$ on the test instance $(x, y)$. We call this the *main task*, as indicated by the subscript of the loss function.

We assume to have training data $(x_1, y_1), \ldots, (x_n, y_n)$ drawn i.i.d. from a distribution $P$. Standard empirical risk minimization corresponds to solving the optimization problem:

$$\min_{\boldsymbol{\theta}} \frac{1}{n} \sum_{i=1}^{n} l_m(x_i, y_i; \boldsymbol{\theta}). \tag{1}$$

Our method requires a *self-supervised auxiliary task* with loss function $l_s(x)$. In this paper, we choose the rotation prediction task (Gidaris et al., 2018), which has been demonstrated to be simple and effective at feature learning for neural networks. The task simply rotates $x$ on the image plane by one of 0, 90, 180 and 270 degrees and have the model predict the angle of rotation as a four-way classification problem. Other self-supervised tasks in section 5 might also be used for our method.

The auxiliary task shares some of the model parameters $\boldsymbol{\theta}_e = (\theta_1, \ldots, \theta_\kappa)$ up to a certain $\kappa \in \{1, \ldots, K\}$. We think of those $\kappa$ layers as a *shared feature extractor*. The auxiliary task uses its own task-specific parameters $\boldsymbol{\theta}_s = (\theta'_{\kappa+1}, \ldots, \theta'_K)$. We call the unshared parameters $\boldsymbol{\theta}_s$ the *self-supervised task branch*, and $\boldsymbol{\theta}_m = (\theta_{\kappa+1}, \ldots, \theta_K)$ the *main task branch*. Pictorially, the joint architecture is a $Y$-structure with a shared bottom and two branches. For our experiments, the self-supervised task branch has the exact same architecture as the main branch, except for the output dimensionality of the last layer due to the different number of classes in the two tasks.

Training is done in the fashion of multi-task learning (Caruana, 1997); the model is trained on both tasks on the same data drawn from $P$. Losses for both tasks are added together, and gradients are taken for the collection of all parameters. The joint training problem is therefore

$$\min_{\boldsymbol{\theta}_e, \boldsymbol{\theta}_m, \boldsymbol{\theta}_s} \frac{1}{n} \sum_{i=1}^{n} l_m(x_i, y_i; \boldsymbol{\theta}_m, \boldsymbol{\theta}_e) + l_s(x_i; \boldsymbol{\theta}_s, \boldsymbol{\theta}_e). \tag{2}$$

Now we describe the standard version of test-time training on a single test instance $x$. Simply put, test-time training finetunes the shared feature extractor $\boldsymbol{\theta}_e$ by minimizing the auxiliary task loss on $x$. This can be formulated as

$$\min_{\boldsymbol{\theta}_e} l_s(x; \boldsymbol{\theta}_s, \boldsymbol{\theta}_e). \tag{3}$$

Denote $\boldsymbol{\theta}_e^*$ the (approximate) minimizer of Equation 3. The model then makes a prediction using the updated parameters $\boldsymbol{\theta}(x) = (\boldsymbol{\theta}_e^*, \boldsymbol{\theta}_m)$. Empirically, the difference is negligible between minimizing Equation 3 over $\boldsymbol{\theta}_e$ versus over both $\boldsymbol{\theta}_e$ and $\boldsymbol{\theta}_s$. Theoretically, there exists a difference only when optimization is done with more than one step of gradient descent.

In the standard version of our method, the optimization problem in Equation 3 is always initialized with parameters $\boldsymbol{\theta} = (\boldsymbol{\theta}_e, \boldsymbol{\theta}_s)$ obtained by minimizing Equation 2 on data from $P$. After making a prediction on $x$, $\boldsymbol{\theta}_e^*$ is discarded. Outside of the standard supervised learning setting, when the test instances arrive online sequentially, the online version of test-time training solves the same optimization problem as in Equation 3 to update the shared feature extractor $\boldsymbol{\theta}_e$. However, on test instance $x_t$, $\boldsymbol{\theta}$ is instead initialized with $\boldsymbol{\theta}(x_{t-1})$ updated on the previous instance $x_{t-1}$. This allows $\boldsymbol{\theta}(x_t)$ to take advantage of the distributional information available in $x_1, \ldots, x_{t-1}$ as well as $x_t$.

Test-time training naturally benefits from standard data augmentation techniques. On each test instance $x$, we perform the exact same set of random transformations as used for data augmentation during training to form a batch only containing these augmented copies of $x$ for test-time training.

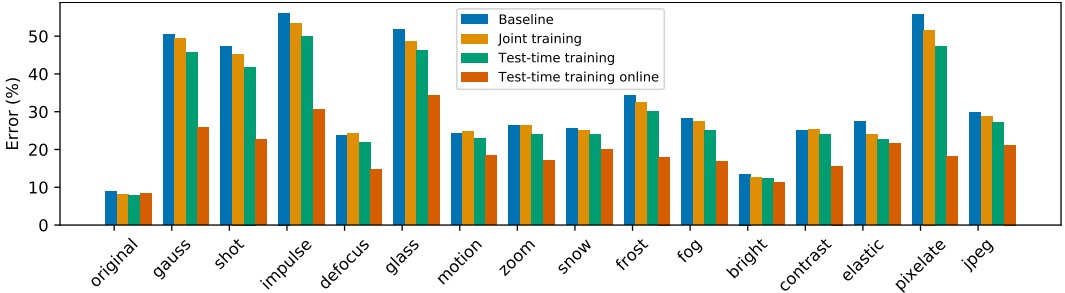

Figure 1: Test error (%) on CIFAR-10-C, level 5. See subsection 3.1 for details.

# 3 EMPIRICAL RESULTS

We experiment with both versions of our method (standard and online) on three kinds of benchmarks for distribution shifts, presented here in the order of visually low to high-level, which is roughly also the order of easy to hard. Our code will be released once the paper is accepted.

**Network details.** Our architecture and hyper-parameters are consistent across all experiments. We use Residual Networks (ResNets) (He et al., 2016b), which are constructed differently for CIFAR-10 [1] (26-layer) and ImageNet [2] (18-layer). ResNets on ImageNet have four groups, each containing convolutional layers with the same number of channels and size of feature maps; our splitting point is the end of the third group. ResNets on CIFAR-10 have three groups; our splitting point is the end of the second group. In addition, Batch Normalization (BN), a popular module in deep networks, is ineffective when training on small batches, for which the estimated batch statistics are less accurate (Ioffe & Szegedy, 2015). This technicality hurts test-time training since each batch only contains (augmented) copies of a single image. Therefore our networks instead use Group Normalization (GN) (Wu & He, 2018), which achieves similar performance as BN on large batches without hurting on small ones. Results with BN are shown in Appendix E for completeness.

**Optimization details.** For Equation 2, optimization hyper-parameters are set to the default [3] in standard practice (Huang et al., 2016; He et al., 2016a). For Equation 3, we use stochastic gradient descent (SGD) with the learning rate set to that of the last epoch during training, which is 0.001 in all our experiments. Following standard practice (He et al., 2018; Liu et al., 2018) known to improve performance when finetuning, we do not use weight decay or momentum. For the standard version, we take ten gradient steps, using batches independently generated by the same image. For online we take only one step. The computational aspects of our method are discussed in Appendix C. Following standard practice, the transformations used for data augmentation are random crop with padding and random horizontal flip for CIFAR-10 (Guo et al., 2017a; Huang et al., 2016), and random resized crop and random horizontal flip for ImageNet (Ioffe & Szegedy, 2015; He et al., 2016a). Specifically, these transformations do not contain information about the distribution shifts.

In all the tables and figures, *baseline* refers to the plain ResNet model (using GN, unless otherwise specified); *joint training* refers to the model jointly trained on both the main task and the self-supervised task, fixed at test-time as in Hendrycks et al. (2019b); *test-time training* refers to the standard version described section 2; and *test-time training online* refers to the online version that does not discard $\theta(x_t)$ for $x_t$ arriving sequentially from the same distribution. Performance for test-time training online is calculated, just like the others, as the average over the entire test set; we always shuffle the test set before test-time training online to avoid ordering artifacts.

---

[1]CIFAR-10 (Krizhevsky & Hinton, 2009) is a standard object recognition dataset with 10 classes of objects in natural scenes. The standard train / test split has 50,000 / 10,000 images, each of size 32-by-32 pixels.

[2]The ImageNet 2012 classification dataset (Russakovsky et al., 2015) for object recognition has images from 1, 000 classes, 1.2 million for training and 50,000 for validation. Following standard practice (He et al., 2016a;b; Huang et al., 2016), the validation set is used as the test set.

[3]Namely, we use stochastic gradient descent (SGD) with weight decay and momentum; learning rate starts at 0.1 and is dropped by a factor of ten at two scheduled milestones, to 0.01 and 0.001.

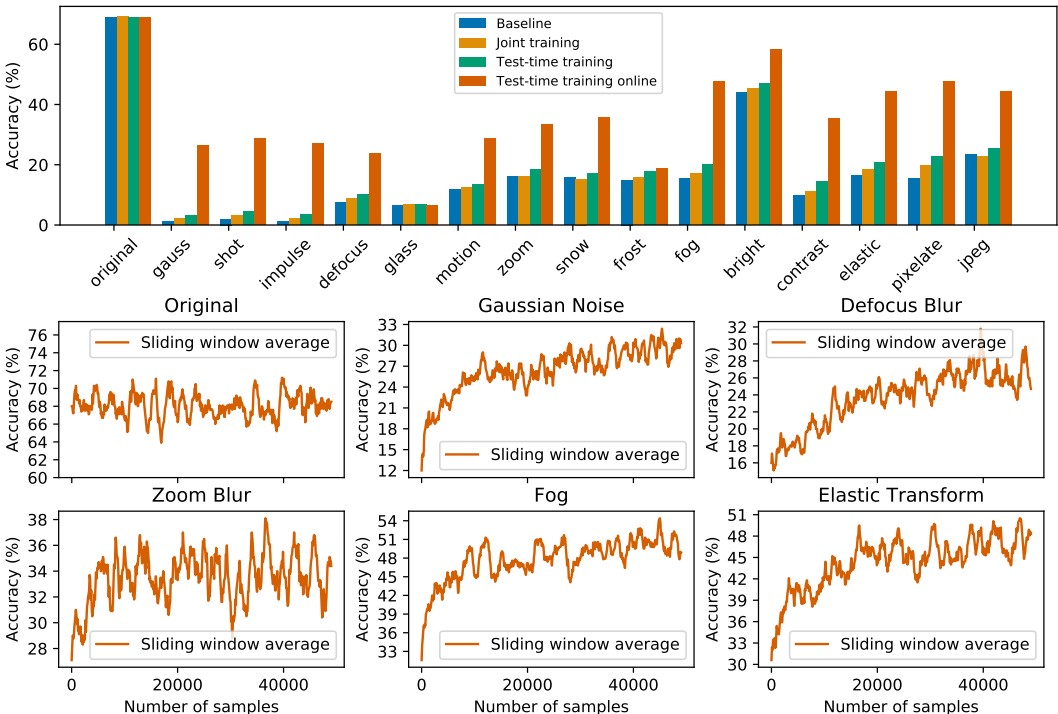

Figure 2: Test accuracy (%) on ImageNet-C, level 5. The lower panel shows the accuracy of the online version as the average over a sliding window of 100 samples; test-time learning online generalizes better as more samples are tested on, without hurting on the original distribution. We use accuracy instead of error here because the baseline performance is very poor with most corruptions. See subsection 3.1 for details.

## 3.1 COMMON CORRUPTIONS

Hendrycks & Dietterich (2019) propose to benchmark robustness of neural networks on 15 types of corruptions from four broad categories: noise, blur, weather and digital. Each corruption type comes in five levels of severity, with level 5 the most severe (details and sample images in Appendix D). The corruptions are algorithmically simulated to mimic real-world corruptions as much as possible on copies of the test set for both CIFAR-10 and ImageNet. According to the authors, training should be done on the original training set, and the diversity of corruption types should make it difficult for any method to work well across the board if it relies too much on corruption specific knowledge.

**CIFAR-10-C.** Our results on the level 5 corruptions (most severe) are shown in Figure 9. Due to space constraints, our results on levels 1-4 are shown in Appendix E. Across all five levels and 15 corruption types, both versions of test-time training always improve over the baseline by a large margin. The standard version of test-time training always improves over joint training, and the online version often improves very significantly (>10%) over joint training and never hurts by more than 0.2%. Specifically, test-time training online contributes >24% on the three noise types and 38% on pixelation. For the seemingly unstable setup of a learning problem that abuses a single image, this kind of consistency is rather surprising.

The baseline ResNet-26 has error 8.9% on the original test set of CIFAR-10. The joint training baseline actually improves performance on the original to 8.1%. Most surprisingly, unlike many other methods that tradeoff original performance with robustness, test-time training further improves on the original test set by 0.2% consistently over many independent trials. This indicates that our method does not choose between specificity and generality.

Separate from our method, it is interesting to note that joint training consistently improves over the baseline, as discovered by Hendrycks et al. (2019b). Hendrycks & Dietterich (2019) have also experimented with various other training methods on this benchmark, and point to Adversarial Logit Pairing (ALP) (Kannan et al., 2018) as the most effective. Results of this additional baseline on

| Accuracy (%) | Airplane | Bird | Car | Dog | Cat | Horse | Ship | Average |
|---|---|---|---|---|---|---|---|---|
| Baseline | 67.9 | 35.8 | 42.6 | 14.7 | 52.0 | 42.0 | 66.7 | **41.4** |
| Joint training | 70.2 | 36.7 | 42.6 | 15.5 | 52.0 | 44.0 | 66.7 | **42.4** |
| Test-time training | 70.2 | 39.2 | 42.6 | 21.6 | 54.7 | 46.0 | 77.8 | **45.2** |
| TTT online | 70.2 | 39.2 | 42.6 | 22.4 | 54.7 | 46.0 | 77.8 | **45.4** |

Table 1: Test accuracy (%) on our video classification dataset, adapted from Shankar et al. (2019). We report accuracy for each class and the average over all samples. See subsection 3.2 for details.

all levels of CIFAR-10-C are shown in Appendix E, along with its implementation details. While surprisingly robust under some of the most severe corruptions (especially the three noise types), ALP incurs a much larger error (by a factor of two) on the original distribution and some corruptions (e.g. all levels of contrast and fog), and hurts performance significantly when the corruptions are not as severe (especially on levels 1-3); this kind of tradeoff is to be expected for methods based on adversarial training, but not test-time training.

**ImageNet-C.** Our results on the level 5 corruptions (most severe) are shown in Figure 2. We use accuracy instead of error for this dataset because the baseline severely underperforms with most corruptions. The general trend is roughly the same as on CIFAR-10-C. Test-time training (standard version) always improves over the baseline and joint training, while the online version only hurts on the original by 0.1% over the baseline, but dramatically improves (by a factor of more than three) on many of the corruption types.

In the lower panel of Figure 2, we visualize how the accuracy (averaged over a sliding window) of the online version changes as more images are tested on. Due to space constraints, we show this plot on the original test set, as well as every third corruption type, following the same order as in the original paper. On the original test set, there is no visible change in performance after updating on the 50,000 samples. With corruptions, accuracy has already risen significantly after 10,000 samples, but is still rising towards the end of the 50,000 samples, indicating room for additional improvements if more samples were available. Without looking at a single label, test-time training online behaves as if we were training on the test set from the appearance of these plots.

## 3.2 VIDEO CLASSIFICATION

The ImageNet Video Classification (VID) dataset was developed by Shankar et al. (2019) from the Video Detection dataset of ILSVRC2015 (Russakovsky et al., 2015), to demonstrate how deep learning models for object recognition trained on ImageNet (still images) fail to adapt well to video frames [4]. Without any modification for videos, we apply our method to VID on top of the same ImageNet model as in the previous subsection. Our results are shown in Table 2. Again, we use accuracy instead of error because the baseline performance is poor.

| Method | Accuracy (%) |
|---|---|
| Baseline | 62.7 |
| Joint training | 63.5 |
| Test-time training | 63.8 |
| TTT online | 64.3 |

Table 2: Test accuracy (%) on VID. See subsection 3.2 for details.

In addition, we take the seven classes in VID that overlap with CIFAR-10, and rescale those video frames to the size of CIFAR-10 images, as a new test set for the model trained on CIFAR-10 in the previous subsection. Again, we apply our method to this dataset without any modification. Our results are shown in Table 1, with a breakdown for each class. Noticing that test-time training does not improve on the airplane class, we inspect some airplane samples, and observe that most of them have black margins on the sides, which provide a trivial hint for the rotation prediction task. In addition, for airplanes captued in the sky, it is often impossible even for humans to tell if an image is rotated. This shows that our method requires the self-supervised task to be both well defined and non-trivial on the new domain.

---

[4] The VID dataset contains 1109 sets of video frames; each set forms a short video clip where all the frames are similar to an anchor frame. Our results are reported on the anchor frames. To map the 1000 ImageNet classes to the 30 VID classes, we use the max-conversion function in Shankar et al. (2019).

### 3.3 CIFAR-10.1: A New Test Set With Unknown Distribution Shifts

CIFAR-10.1 (Recht et al., 2018) is a new test set of size 2000 modeled after CIFAR-10, with the exact same classes and image dimensionality, following the dataset creation process documented by the original CIFAR-10 paper as closely as possible. The purpose is to investigate the distribution shifts present between the two test sets, and the effect on object recognition. All models tested by the authors suffer a large performance drop on CIFAR-10.1 comparing to CIFAR-10, even though there is no human noticable difference, and both have the same human accuracy. This demonstrates how insidious and ubiquitous distribution shifts are, even when researchers strive to minimize them.

The distribution shifts from CIFAR-10 to CIFAR-10.1 pose an extremely difficult problem, and nobody has made a successful attempt to improve the performance of an existing model on this new test set, probably because 1) Researchers cannot even identify the distribution shifts, let alone describe them with mathematics. 2) The samples in CIFAR-10.1 are only revealed at test-time; and even if revealed during training, the distribution shifts are too subtle, and the sample size is too small, for domain adaptation algorithms (Recht et al., 2018).

| Method | Error (%) |
| --- | --- |
| Baseline | 17.4 |
| Joint training | 16.7 |
| Test-time training | 15.9 |

Table 3: Test error (%) on CIFAR-10.1. See subsection 3.3 for details.

On the original CIFAR-10 test set, our baseline has error 8.9%, and with joint training 8.1%; comparing to the first two rows of Table 3, both suffer the typical performance drop (by a factor of two). Test-time training yields an improvement of 0.8% (relative improvement of 4.8%) over joint training. We recognize that this improvement is still small comparing to the performance drop, but see it as an encouraging first step for this very difficult problem.

## 4 Towards Understanding Test-time Training

This section contains our preliminary study of when and why test-time training is expected to work. For convex models, we prove that positive gradient correlation between the loss functions leads to better performance on the main task after test-time training. Equipped with this insight, we then empirically demonstrate that gradient correlation governs the success of test-time training on the deep learning model discussed in Section 3.

Before stating our main theoretical result, we first illustrate the general intuition with a toy model. Consider a regression problem where $x \in \mathbb{R}^d$ denotes the input, $y_1 \in \mathbb{R}$ denotes the label, and the objective is the square loss $(\hat{y} - y_1)^2/2$ for a prediction $\hat{y}$. Consider a two layer linear network parametrized by $\boldsymbol{A} \in \mathbb{R}^{h \times d}$ and $\boldsymbol{v} \in \mathbb{R}^h$ (where $h$ stands for the hidden dimension). The prediction according to this model is $\hat{y} = \boldsymbol{v}^\top \boldsymbol{A} x$, and the main task loss is

$$l_m(x, y_1; \boldsymbol{A}, \boldsymbol{v}) = \frac{1}{2} \left( y_1 - \boldsymbol{v}^\top \boldsymbol{A} x \right)^2. \tag{4}$$

In addition, consider a self-supervised regression task that also uses the square loss and automatically generates a label $y_s$ for $x$. Let the self-supervised head be parameterized by $\boldsymbol{w} \in \mathbb{R}^h$. Then the self-supervised task loss is

$$l_s(x, y_2; \boldsymbol{A}, \boldsymbol{w}) = \frac{1}{2} \left( y_2 - \boldsymbol{w}^\top \boldsymbol{A} x \right)^2. \tag{5}$$

Now we apply test-time training to update the shared feature extractor $\boldsymbol{A}$ by one step of gradient descent on $l_s$, which we can compute with $y_2$ known. This gives us

$$\boldsymbol{A}' \leftarrow \boldsymbol{A} - \eta \left( y_2 - \boldsymbol{w}^\top \boldsymbol{A} x \right) \left( -\boldsymbol{w} x^\top \right), \tag{6}$$

where $\boldsymbol{A}'$ is the updated matrix and $\eta$ is the learning rate. If we set $\eta = \eta^*$ where

$$\eta^* = \frac{y_1 - \boldsymbol{v}^\top \boldsymbol{A} x}{\left( y_2 - \boldsymbol{w}^\top \boldsymbol{A} x \right) \boldsymbol{v}^\top \boldsymbol{w} x^\top x}, \tag{7}$$

then with some simple algebra, it is easy to see that the main task loss $l_m(x, y_1; \boldsymbol{A}', \boldsymbol{v}) = 0$. Concretely, test-time training drives the main task loss down to zero with a single gradient step for a carefully chosen learning rate. In practice, this learning rate is unknown since it depends on the unknown $y_1$. However, since our model is convex, as long as $\eta^*$ is positive, it suffices to set $\eta$ to

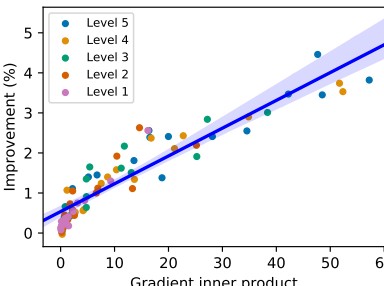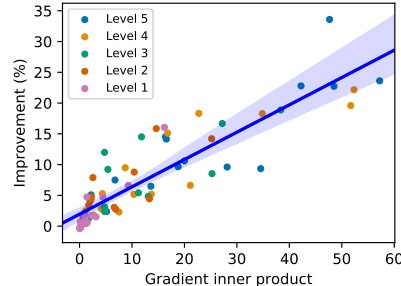

Figure 3: Scatter plot of the inner product between the gradients (on the shared feature extractor $\boldsymbol{\theta}_e$) of the main task $l_m$ and the self-supervised task $l_e$, and the improvement in test error (%) from test-time training, for the standard (left) and online (right) version. Each point is the average over a test set, and each scatter plot has 75 test sets, from all 15 types of corruptions over five levels as described in subsection 3.1. The blue lines and bands are the best linear fits and the 99% confidence intervals. The linear correlation coefficients are 0.93 and 0.89 respectively, indicating strong positive correlation between the two quantities, as suggested by Theorem 1.

be a small positive constant (see Lemma 1). If $x \neq 0$, one sufficient condition for $\eta^*$ to be positive (when neither loss is zero) is to have

$$\text{sign}\left(y_1 - \boldsymbol{v}^\top \boldsymbol{A} x\right) = \text{sign}\left(y_2 - \boldsymbol{w}^\top \boldsymbol{A} x\right) \quad \text{and} \quad \boldsymbol{v}^\top \boldsymbol{w} > 0 . \tag{8}$$

For our toy model, both parts in Equation 8 have an intuition interpretation. The first part says that the mistakes should be correlated, in the sense that predictions from both tasks are mistaken in the same direction. The second part, $v^\top \boldsymbol{w} > 0$, says that the decision boundaries on the feature space should be correlated. In fact, these two parts hold iff $\langle \nabla l_m(\boldsymbol{A}), \nabla l_s(\boldsymbol{A}) \rangle > 0$ (see Lemma 2). To summarize, if the gradients have positive correlation, test-time training is guaranteed to reduce the main task loss. Our main theoretical result extends this to general smooth and convex loss functions.

**Theorem 1.** *Let $l_m(x, y; \boldsymbol{\theta})$ denote the main task loss on test instance $x, y$ with parameters $\boldsymbol{\theta}$, and $l_s(x; \boldsymbol{\theta})$ the self-supervised task loss that only depends on $x$ [5]. Assume that for all $x, y$, $l_m(x, y; \boldsymbol{\theta})$ is differentiable, convex and $\beta$-smooth in $\boldsymbol{\theta}$, and both $\|\nabla l_m(x, y; \boldsymbol{\theta})\|, \|\nabla l_s(x, \boldsymbol{\theta})\| \leq G$ for all $\boldsymbol{\theta}$. With a fixed learning rate $\eta = \frac{\epsilon}{\beta G^2}$, for every $x, y$ such that*

$$\langle \nabla l_m(x, y; \boldsymbol{\theta}), \nabla l_s(x; \boldsymbol{\theta}) \rangle > \epsilon, \tag{9}$$

*we have*

$$l_m(x, y; \boldsymbol{\theta}) > l_m(x, y; \boldsymbol{\theta}(x)), \tag{10}$$

*where $\boldsymbol{\theta}(x) = \boldsymbol{\theta} - \eta \nabla l_s(x; \boldsymbol{\theta})$ i.e. test-time training with one step of gradient descent.*

The proof is in Appendix B.2. Theorem 1 reveals gradient correlation as a determining factor of the success of test-time training in the smooth and convex case. In Figure 3, we empirically show that our insight also holds for non-convex loss functions, on the deep learning model and across the diverse set of corruptions considered in section 3; stronger gradient correlation clearly indicates higher performance improvements over the baseline.

## 5 RELATED WORK

Our work has been influenced by the successes and limitations of many related fields. Each of these fields contains maybe hundreds of interesting works, which we unfortunately do not have enough time and space to acknowledge in this draft. We apologize for the omissions and are happy to include additional citations upon request.

**Learning on test instances.** Concurrent work Bau et al. (2019) improves photo manipulation with a generative adversarial network, by adapting its image prior to the statistics of the input image. Shocher et al. (2018) performs super-resolution on an image, by learning to recover the original

---

[5]Because the main task branch and the self-supervised branch are kept fixed at test-time, we do not explicitly describe their parameters, and include them implicitly in the loss functions.

image itself from its downsampled version. An older work, Jain & Learned-Miller (2011) improves Viola-Jones (Viola et al., 2001) for face detection, by bootstrapping the more difficult faces in an image from the easier ones with high confidence in the same image. The online version of our algorithm is inspired by Mullapudi et al. (2018), which makes video segmentation more efficient by using a student model that learns online from a teacher model. The idea of online updates has also been used in Kalal et al. (2011) for tracking and detection. Zhu et al. (2019), a concurrent work in echocardiography, improves the deep learning model that tracks myocardial motion and cardiac blood flow with sequential updates. Lastly, we share the philosophy of transductive learning (Vapnik, 2013; Gammerman et al., 1998), but have little in common with their classical algorithms; concurrent work Tripuraneni & Mackey (2019) theoretically explores this for linear prediction.

**Adversarial robustness** studies the robust risk: $R_{P,\Delta}(\boldsymbol{\theta}) = \mathbb{E}_{x,y \sim P} \max_{\delta \in \Delta} l(x+\delta, y; \boldsymbol{\theta})$ where $l$ is some loss function, and $\Delta$ is the set of perturbations; $\Delta$ is often chosen as the $L_p$ ball, for $p \in \{1, 2, \infty\}$. Many popular algorithms formulate and solve this as a robust optimization problem (Goodfellow et al., 2014; Madry et al., 2017; Sinha et al., 2017; Raghunathan et al., 2018; Wong & Kolter, 2017; Croce et al., 2018), and the most well known technique is adversarial training. Another line of work is based on randomized smoothing (Cohen et al., 2019; Salman et al., 2019), while some other approaches such as input transformations (Guo et al., 2017b; Song et al., 2017) are shown to be less effective (Athalye et al., 2018). There are two main problems in this field. First, all the approaches mentioned above can be seen as *smoothing* the decision boundary. This establishes a theoretical tradeoff between accuracy and robustness (Tsipras et al., 2018; Zhang et al., 2019), which we also observe empirically with our adversarial training baseline in Section 3. Intuitively, the more diverse $\Delta$ is, the less effective this *one-boundary-fits-all* approach can be for a particular element of $\Delta$. Second, adversarial methods rely heavily on the mathematical structure of $\Delta$, which might not accurately model perturbations in the real world. Therefore, generalization remains hard outside of the $\Delta$ we know in advance or can mathematically model, especially for non-adversarial distribution shifts. Empirically, Kang et al. (2019) shows that robustness for one $\Delta$ might not transfer to another, and training on the $L_\infty$ ball actually hurts robustness on the $L_1$ ball.

**Non-adversarial robustness** studies the effect of corruptions, perturbations, out-of-distribution examples, and real-world distribution shifts (Hendrycks et al., 2019a;b; 2018; Hendrycks & Gimpel, 2016). Geirhos et al. (2018) show that training on images corrupted by Gaussian noise makes deep learning models recover super-human performance on this particular noise type, but cannot improve performance on another those corrupted by another noise type e.g. salt-and-pepper noise.

**Unsupervised domain adaptation** (a.k.a. transfer learning) studies the problem of distribution shift (from $P$ to $Q$), when unlabled data from $Q$ is available at training-time (Tzeng et al., 2017; Ganin et al., 2016; Gong et al., 2012; Long et al., 2016; Chen et al., 2018; 2011; Hoffman et al., 2017; Csurka, 2017; Long et al., 2015). We are inspired by this very active and successful community, especially (Sun et al., 2019), and believe that progress in this community can motivate new algorithms in the framework of test-time learning. Our update rule can be viewed as performing *one-sample unsupervised domain adaptation* on the fly [6]. On the other hand, test-time learning comes from realizing the limitations of the unsupervised domain adaptation setting, that outside of the specific target distribution where data is available for training, generalization is still elusive. Previous works make the source and target distributions broader with multiple and evolving sources and targets without fundamentally address this problem (Hoffman et al., 2018; 2012; 2014).

**Self-supervised learning** studies how to create labels from the data, by designing ingenious tasks that contain semantic information without human annotations, such as context prediction (Doersch et al., 2015), solving jigsaw puzzles (Noroozi & Favaro, 2016), colorization (Larsson et al., 2017; Zhang et al., 2016), noise prediction (Bojanowski & Joulin, 2017), and feature clustering (Caron et al., 2018). Self-supervision has also been used on videos (Wang & Gupta, 2015; Wang et al., 2019). Particularly relevant to our work, Asano et al. (2019) shows that self-supervised learning on only a single image, surprisingly, can produce low-level features that generalize well. In addition,

---

[6]Note that typical unsupervised domain adaptation algorithms,such as those based on distributional discrepancy, adversarial learning, co-training and generative modeling, might not work in our framework because the concept of a target distribution, which has been so deeply rooted and heavily relied on, becomes ill-defined when there is only one sample from the target domain.

Hendrycks et al. (2019b) proposes that jointly training a main task and a self-supervised task (our joint training baseline in section 3) can improve robustness of the main task. The same idea is used in few-shot learning (Su et al., 2019), and domain generalization (Carlucci et al., 2019).

**Domain generalization** studies when a meta distribution generates multiple environment distributions, some of which are available during training (source), while others are used for testing (target) (Li et al., 2018; Shankar et al., 2018; Muandet et al., 2013; Balaji et al., 2018; Ghifary et al., 2015; Motiian et al., 2017; Li et al., 2017a; Gan et al., 2016). With only a few environments, information on the meta distribution is often too scarce to be helpful, and with many environments, we are back to the i.i.d. setting where each environment can be seen as a sample, and a strong baseline is to simply train on all the environments (Li et al., 2019). The setting of domain generalization is limited by the inherent tradeoff between specificity and generality of a fixed decision boundary, and the fact that generalization is again elusive outside of the meta distribution i.e. the actual $P$.

**Continual learning** (a.k.a. learning without forgetting) studies when a model is made to learn a sequence of tasks, and not forget about the task at the beginning (Li & Hoiem, 2017; Lopez-Paz & Ranzato, 2017; Kirkpatrick et al., 2017; Santoro et al., 2016). Test-time training does not care about forgetting the past test instances since they have already been evaluated on; and if a past instance comes up by any chance, it would go through test-time training again. In addition, the impact of forgetting the training set is minimal, because both tasks have been jointly trained. This is in contrast to continual learning, when the tasks are trained one-by-one from scratch.

**Few (one)-shot learning** studies extremely small training sets (maybe for some categories) (Snell et al., 2017; Vinyals et al., 2016; Fei-Fei et al., 2006; Ravi & Larochelle, 2016; Li et al., 2017b; Finn et al., 2017; Gidaris & Komodakis, 2018). Our update rule can be viewed as performing *one-shot self-supervised learning* and can potentially be improved by progress in few-shot learning.

**Online learning** (a.k.a. online optimization) is a well-studied area of learning theory (Shalev-Shwartz et al., 2012; Hazan et al., 2016). The basic setting repeats the following: receive $x_t$, predict $\hat{y}_t$, receive $y_t$ from a worst-case oracle and learn. Final performance is evaluated using the regret, colloquially how much worse than the best fixed model in hindsight. It is easy to see how our setting differs, even for the online version. We learn before predicting $\hat{y}_t$, but never receive any $y_t$ that is evaluated on, thus do not need to consider the worst-case oracle or the regret.

## 6 CONCLUSION

The idea of test-time training also makes sense for other tasks such as segmentation and detection, and in other fields such as speech recognition and natural language processing. More generally, we hope this paper can encourage researchers to abandon the self-imposed constraint of a fixed decision boundary at test-time, or even altogether, the artificial divison between training and testing.

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

## A    INFORMAL DISCUSSION ON OUR VARIABLE DECISION BOUNDARY

In section 1, we claim that in traditional supervised learning $\boldsymbol{\theta}$ gives a fixed decision boundary, while our $\boldsymbol{\theta}$ gives a variable decision boundary. Here we informally discuss this claim.

Denote the input space $\mathcal{X}$ and output space $\mathcal{Y}$. A decision boundary is simply a mapping $f : \mathcal{X} \to \mathcal{Y}$. Let $\Theta$ be a model class e.g $\mathbb{R}^d$. Now consider a family of parametrized functions $g_{\boldsymbol{\theta}} : \mathcal{X} \to \mathcal{Y}$, where $\boldsymbol{\theta} \in \Theta$. In the context of deep learning, $g$ is the neural network architecture and $\boldsymbol{\theta}$ contains the parameters. We say that $f$ is a fixed decision boundary w.r.t. $g$ and $\Theta$ if there exists $\boldsymbol{\theta} \in \Theta$ s.t. $f(x) = g_{\boldsymbol{\theta}}(x)$ for every $x \in \mathcal{X}$, and a variable decision boundary if for every $x \in \mathcal{X}$, there exists $\boldsymbol{\theta} \in \Theta$ s.t. $f(x) = g_{\boldsymbol{\theta}}(x)$. Note how selection of $\boldsymbol{\theta}$ can depend on $x$ for a variable decision boundary, and cannot for a fixed one. It is then trivial to verify that our claim is true under those definitions.

A critical reader might say that with an arbitrarily large model class, can't every decision boundary be fixed? Yes, but this is not the end of the story. Let $d = \dim(\mathcal{X}) \times \dim(\mathcal{Y})$, and consider the enormous model class $\Theta' = \mathbb{R}^d$ which is capable of representing all possible mappings between $\mathcal{X}$ and $\mathcal{Y}$. Let $g'_{\boldsymbol{\theta}'}$ simply be the mapping represented by $\boldsymbol{\theta}' \in \Theta'$. A variable decision boundary w.r.t. $g$ and $\Theta$ then indeed must be a fixed decision boundary w.r.t. $g'$ and $\Theta'$, but we would like to note two things. First, without any prior knowledge, generalization in $\Theta'$ is impossible with any finite amount of training data; reasoning about $g'$ and $\Theta'$ is most likely not productive from an algorithmic point of view, and the concept of a variable decision boundary is to avoid such reasoning. Second, selecting $\boldsymbol{\theta}$ based on $x$ for a variable decision boundary can be thought of as "training" on all points $x \in \mathbb{R}^d$; however, "training" only happens when necessary, for the $x$ that it actually encounters.

Altogether, the concept of a variable decison boundary is different from what can be described by traditional learning theory. A formal discussion is beyond the scope of this paper.

## B    PROOFS

Here we prove the theoretical results covered in section 4.

### B.1    PROOFS FOR THE TOY PROBLEM

The following setting applies to the two lemmas; this is simply the setting of our toy problem, reproduced here for ease of reference. Consider a two layer linear network parametrized by $\boldsymbol{A} \in \mathbb{R}^{h \times d}$ (shared) and $\boldsymbol{v}, \boldsymbol{w} \in \mathbb{R}^h$ (fixed) for the two heads, respectively. Denote $x \in \mathbb{R}^d$ the input and $y_1, y_2 \in \mathbb{R}$ the labels for the two tasks, respectively. For the main task loss

$$l_m(\boldsymbol{A}; \boldsymbol{v}) = \frac{1}{2} \left(y_1 - \boldsymbol{v}^\top \boldsymbol{A} x\right)^2, \tag{11}$$

and the self-supervised task loss

$$l_s(\boldsymbol{A}; \boldsymbol{w}) = \frac{1}{2} \left(y_2 - \boldsymbol{w}^\top \boldsymbol{A} x\right)^2, \tag{12}$$

test-time learning yields an updated matrix

$$\boldsymbol{A}' \leftarrow \boldsymbol{A} - \eta \left(y_2 - \boldsymbol{w}^\top \boldsymbol{A} x\right) \left(-\boldsymbol{w} x^\top\right), \tag{13}$$

where $\eta$ is the learning rate.

**Lemma 1.** *Following the exposition of section 4, denote*

$$\eta^* = \frac{(y_1 - v^\top A x)}{(y_2 - w^\top A x) v^\top w x^\top x}. \tag{14}$$

*Assume $\eta^* \in [\epsilon, \infty)$ for some $\epsilon > 0$. Then for any $\eta \in (0, \epsilon]$, we are guaranteed an improvement on the main loss i.e. $l_m(\boldsymbol{A}') < l_m(\boldsymbol{A})$.*

*Proof.* From the exposition of section 4, we know that

$$l_m(\boldsymbol{A} - \eta^* \nabla l_s \boldsymbol{A})) = 0,$$

which can also be derived from simple algebra. Then by convexity, we have

$$l_m \left( \boldsymbol{A} + \eta \nabla l_s(\boldsymbol{A}) \right) = l_m \left( \left( 1 - \frac{\eta}{\eta^*} \right) \boldsymbol{A} + \frac{\eta}{\eta^*} (\boldsymbol{A} - \eta^* \nabla l_s(\boldsymbol{A})) \right) \tag{15}$$

$$\leq \left( 1 - \frac{\eta}{\eta^*} \right) l_m(\boldsymbol{A}) + 0 \tag{16}$$

$$\leq \left( 1 - \frac{\eta}{\epsilon} \right) l_m(\boldsymbol{A}) \tag{17}$$

$$< l_m(\boldsymbol{A}), \tag{18}$$

where the last inequality uses the assumption that $l_m(\boldsymbol{A}) > 0$, which holds because $\eta^* > 0$.

**Lemma 2.** *Define $\langle \boldsymbol{U}, \boldsymbol{V} \rangle = vec\,(\boldsymbol{U})^\top vec\,(\boldsymbol{V})$ i.e. the Frobenious inner product, then*

$$\mathrm{sign}\,(\eta^*) = \mathrm{sign}\,(\langle \nabla l_m(\boldsymbol{A}), \nabla l_s(\boldsymbol{A}) \rangle). \tag{19}$$

*Proof.* By simple algebra,

$$\langle \nabla l_m(\boldsymbol{A}), \nabla l_s(\boldsymbol{A}) \rangle = \langle \left( y_1 - \boldsymbol{v}^\top \boldsymbol{A} x \right) \left( -\boldsymbol{v} x^\top \right), \left( y_2 - \boldsymbol{w}^\top \boldsymbol{A} x \right) \left( -\boldsymbol{w} x^\top \right) \rangle \tag{20}$$

$$= \left( y_1 - \boldsymbol{v}^\top \boldsymbol{A} x \right) \left( y_2 - \boldsymbol{w}^\top \boldsymbol{A} x \right) \mathrm{Tr}\,\left( x \boldsymbol{v}^\top \boldsymbol{w} x^\top \right) \tag{21}$$

$$= \left( y_1 - \boldsymbol{v}^\top \boldsymbol{A} x \right) \left( y_2 - \boldsymbol{w}^\top \boldsymbol{A} x \right) \boldsymbol{v}^\top \boldsymbol{w} x^\top x, \tag{22}$$

which has the same sign as $\eta^*$.

### B.2 PROOF OF THEOREM 1

For any $\eta$, by smoothness and convexity,

$$l_m(x, y; \boldsymbol{\theta}(x)) = l_m(x, y; \boldsymbol{\theta} - \eta \nabla l_s(x; \boldsymbol{\theta})) \tag{23}$$

$$\leq l_m(x, y; \boldsymbol{\theta}) + \eta \langle \nabla l_m(x, y; \boldsymbol{\theta}), \nabla l_s(x, \boldsymbol{\theta}) \rangle + \frac{\eta^2 \beta}{2} \left\| \nabla l_s(x; \boldsymbol{\theta}) \right\|^2. \tag{24}$$

Denote

$$\eta^* = \frac{\langle \nabla l_m(x, y; \boldsymbol{\theta}), \nabla l_s(x, \boldsymbol{\theta}) \rangle}{\beta \left\| \nabla l_s(x; \boldsymbol{\theta}) \right\|^2}.$$

Then Equation 23 becomes

$$l_m(x, y; \boldsymbol{\theta} - \eta^* \nabla l_s(x; \boldsymbol{\theta})) \leq l_m(x, y; \boldsymbol{\theta}) - \frac{\langle \nabla l_m(x, y; \boldsymbol{\theta}), \nabla l_s(x, \boldsymbol{\theta}) \rangle^2}{2\beta \left\| \nabla l_s(x; \boldsymbol{\theta}) \right\|^2}. \tag{25}$$

And by our assumptions on the gradient norm and gradient inner product,

$$l_m(x, y; \boldsymbol{\theta}) - l_m(x, y; \boldsymbol{\theta} - \eta^* \nabla l_s(x; \boldsymbol{\theta})) \geq \frac{\epsilon^2}{2\beta G^2}. \tag{26}$$

Because we cannot observe $\eta^*$ in practice, we instead use a fixed learning rate $\eta = \frac{\epsilon}{\beta G^2}$, as stated in Theorem 1. Now we argue that this fixed learning rate still improves performance on the main task.

By our assumptions, $\eta^* \geq \frac{\epsilon}{\beta G^2}$, so $\eta \in (0, \eta^*]$. Denote $\boldsymbol{g} = \nabla l_s(x; \boldsymbol{\theta})$, then by convexity of $l_m$,

$$l_m(x, y; \boldsymbol{\theta}(x)) = l_m(x, y; \boldsymbol{\theta} - \eta \boldsymbol{g}) \tag{27}$$

$$= l_m \left( x, y; \left( 1 - \frac{\eta}{\eta^*} \right) \boldsymbol{\theta} + \frac{\eta}{\eta^*} (\boldsymbol{\theta} - \eta^* \boldsymbol{g}) \right) \tag{28}$$

$$\leq \left( 1 - \frac{\eta}{\eta^*} \right) l_m(x, y; \boldsymbol{\theta}) + \frac{\eta}{\eta^*} l_m(x, y; \boldsymbol{\theta} - \eta^* \boldsymbol{g}) \tag{29}$$

Combining with Equation 26, we have

$$l_m(x, y; \boldsymbol{\theta}(x)) \leq \left( 1 - \frac{\eta}{\eta^*} \right) l_m(x, y; \boldsymbol{\theta}) + \frac{\eta}{\eta^*} \left( l_m(x, y; \boldsymbol{\theta}) - \frac{\epsilon^2}{2\beta G^2} \right) \tag{30}$$

$$= l_m(x, y; \boldsymbol{\theta}) - \frac{\eta}{\eta^*} \frac{\epsilon^2}{2\beta G^2} \tag{31}$$

Since $\eta/\eta^* > 0$, we have shown that

$$l_m(x, y; \boldsymbol{\theta}) - l_m(x, y; \boldsymbol{\theta}(x)) > 0. \tag{32}$$

## C    COMPUTATIONAL ASPECTS OF OUR METHOD

At test-time, our method is $2\times$`batch_size`$\times$`number_of_iterations` times slower than regular testing, which only performs a single forward pass for each sample. As the first work on test-time learning, this paper is not as concerned about computational efficiency as improving robustness, but here we provide two potential solutions that might be useful, but have not been thoroughly verified. The first is to use the thresholding trick on $l_s$, introduced as a solution for the small batches problem in section 2. For the models considered in our experiments, roughly $80\%$ of the test instances fall below the threshold, so test-time learning can only be performed on the other $20\%$ without much effect on performance, because those $20\%$ contain most of the samples with wrong predictions. The second is to reduce the `number_of_iterations` of test-time updates. For the online version, the `number_of_iterations` is already 1, so there is nothing to do. For the standard version, we have done some preliminary experiments setting `number_of_iterations` to 1 (instead of 10) and learning rate to 0.01 (instead of 0.001), and observing results almost as good as the standard hyper-parameter setting. A more in depth discussion on efficiency is left for future works, which might, during training, explicitly make the model amenable to fast updates.

## D    SAMPLE IMAGES FROM THE COMMON CORRUPTIONS BENCHMARK

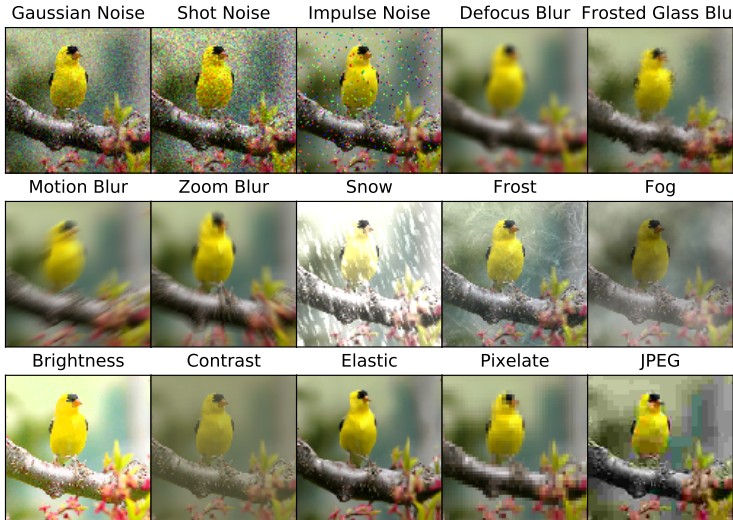

Figure 4: Sample images from the Common Corruptions Benchmark, taken from the original paper by Hendrycks & Dietterich (2019).

## E    ADDITIONAL RESULTS ON THE COMMON CORRUPTIONS DATASET

For table aethetics, we use the following abbreviations: B for baseline, JT for joint training, TTT for test-time training standard version, and TTTO for test-time training online version.

We have also abbreviated the names of the corruptions. The full names are, in order: original test set, Gaussian noise, shot noise, impulse noise, defocus blur, glass blue, motion blur, zoom blur, snow, frost, fog, brightness, contrast, elastic transformation, pixelation, and JPEG compression.

### E.1    RESULTS USING BATCH NORMALIZATION

As discussed in section 3, Batch Normalization (BN) is ineffective for small batches, which are the inputs for test-time training (both standard and online version) since there is only one sample available when forming each batch; therefore, our main results are based on a ResNet using Group Normalization (GN). Here we provide results of our method on CIFAR-10-C level 5, with a ResNet using Batch Normalization (BN). These results are meant to be merely a point of reference for the curious readers, instead of our technical contributions.

In the early stage of this project, we have experimented with two potential solutions to the small batches problem with BN. The naive solution is to fix the BN layers during test-time training. but this diminishes the performance gains since there are fewer shared parameters. The better solution, adopted for the results below, is hard example mining: instead of updating on all inputs, we only update on inputs that incur large self-supervised task loss $l_s$, where the large improvements might counter the negative effects of inaccurate statistics.

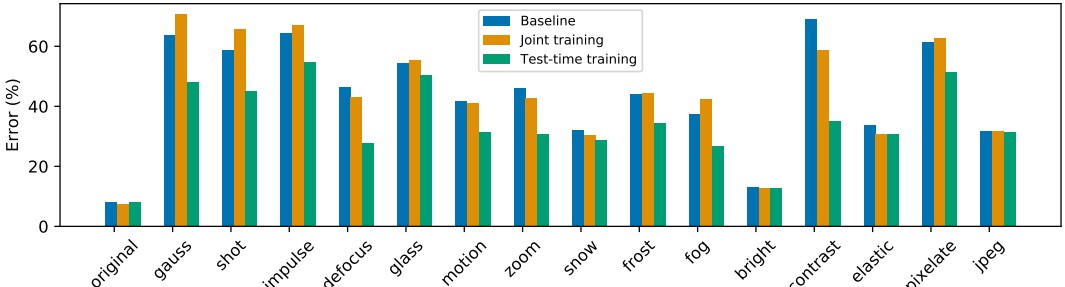

Figure 5: Test error (%) on CIFAR-10-C, level 5, ResNet-26 with Batch Normalization.

| | orig | gauss | shot | impul | defoc | glass | motn | zoom | snow | frost | fog | brit | contr | elast | pixel | jpeg |
|---|---|---|---|---|---|---|---|---|---|---|---|---|---|---|---|---|
| B | 7.9 | 63.9 | 58.8 | 64.3 | 46.3 | 54.6 | 41.6 | 45.9 | 31.9 | 44.0 | 37.5 | 13.0 | 69.2 | 33.8 | 61.4 | 31.7 |
| JT | 7.5 | 70.7 | 65.6 | 67.2 | 43.1 | 55.4 | 40.9 | 42.7 | 30.3 | 44.5 | 42.5 | 12.7 | 58.6 | 30.7 | 62.6 | 31.9 |
| TTT | 7.9 | 47.9 | 45.2 | 54.8 | 27.6 | 50.4 | 31.5 | 30.9 | 28.7 | 34.3 | 26.9 | 12.6 | 35.2 | 30.6 | 51.2 | 31.3 |

Table 4: Test error (%) on CIFAR-10-C, level 5, ResNet-26 with Batch Normalization.

Test-time training (standard version) is still very effective with BN. In fact, some of the improvements are quite dramatic, such as on contrast (34%), defocus blue (18%) and Gaussian noise (22% comparing to joint-training, and 16% comparing to the baseline). Performance on the original distribution is still almost the same, and the original error with BN is in fact slightly lower than with GN, and takes half as many epochs to converge.

We did not further experiment with BN because of two reasons: 1) The online version does not work with BN, because the problem with inaccurate batch statistics is exacerbated when training online for many (e.g. 10000) steps. 2) The baseline error for almost every corruption type is significantly higher with BN than with GN. Although unrelated to the main idea of our paper, we make the interesting note that *GN significantly improves model robustness*.

## E.2  ADDITIONAL BASELINE: ADVERSARIAL LOGIT PAIRING

As discussed in subsection 3.1, Hendrycks & Dietterich (2019) point to Adversarial Logit Pairing (ALP) (Kannan et al., 2018) as an effective method for improving model robustness to corruptions and perturbations, even though it was designed to defend against adversarial attacks. We take ALP as an additional baseline on all benchmarks based on CIFAR-10 (using GN), following the training procedure in Kannan et al. (2018) and their recommended hyper-parameters. The implementation of the adversarial attack comes from the codebase of Ding et al. (2019). We did not run ALP on ImageNet because the two papers we reference for this method, Kannan et al. (2018) and Hendrycks & Dietterich (2019), did not run on ImageNet or make any claim or recommendation.

### E.3 RESULTS ON CIFAR-10-C AND IMAGENET-C, LEVEL 5

The following two tables correspond to the bar plots in section 3.

|      | orig | gauss | shot | impul | defoc | glass | motn | zoom | snow | frost | fog | brit | contr | elast | pixel | jpeg |
|------|------|-------|------|-------|-------|-------|------|------|------|-------|-----|------|-------|-------|-------|------|
| B    | 8.9  | 50.5  | 47.2 | 56.1  | 23.7  | 51.7  | 24.3 | 26.3 | 25.6 | 34.4  | 28.1 | 13.5 | 25.0  | 27.4  | 55.8  | 29.8 |
| JT   | 8.1  | 49.4  | 45.3 | 53.4  | 24.2  | 48.5  | 24.8 | 26.4 | 25.0 | 32.5  | 27.5 | 12.6 | 25.3  | 24.0  | 51.6  | 28.7 |
| TTT  | 7.9  | 45.6  | 41.8 | 50.0  | 21.8  | 46.1  | 23.0 | 23.9 | 23.9 | 30.0  | 25.1 | 12.2 | 23.9  | 22.6  | 47.2  | 27.2 |
| TTTO | 8.2  | 25.8  | 22.6 | 30.6  | 14.6  | 34.4  | 18.3 | 17.1 | 20.0 | 18.0  | 16.9 | 11.2 | 15.6  | 21.6  | 18.1  | 21.2 |
| ALP  | 16.5 | 22.7  | 22.9 | 28.3  | 25.0  | 25.6  | 27.4 | 23.1 | 25.2 | 27.2  | 64.8 | 21.7 | 73.6  | 23.0  | 20.2  | 18.9 |

Table 5: Test error (%) on CIFAR-10-C, level 5, ResNet-26.

|      | orig | gauss | shot | impul | defoc | glass | motn | zoom | snow | frost | fog | brit | contr | elast | pixel | jpeg |
|------|------|-------|------|-------|-------|-------|------|------|------|-------|-----|------|-------|-------|-------|------|
| B    | 68.9 | 1.3   | 2.0  | 1.3   | 7.5   | 6.6   | 11.8 | 16.2 | 15.7 | 14.9  | 15.3 | 43.9 | 9.7   | 16.5  | 15.3  | 23.4 |
| JT   | 69.1 | 2.1   | 3.1  | 2.1   | 8.7   | 6.7   | 12.3 | 16.0 | 15.3 | 15.8  | 17.0 | 45.3 | 11.0  | 18.4  | 19.7  | 22.9 |
| TTT  | 69.0 | 3.1   | 4.5  | 3.5   | 10.1  | 6.8   | 13.5 | 18.5 | 17.1 | 17.9  | 20.0 | 47.0 | 14.4  | 20.9  | 22.8  | 25.3 |
| TTTO | 68.8 | 26.3  | 28.6 | 26.9  | 23.7  | 6.6   | 28.7 | 33.4 | 35.6 | 18.7  | 47.6 | 58.3 | 35.3  | 44.3  | 47.8  | 44.3 |

Table 6: Test accuracy (%) on ImageNet-C, level 5, ResNet-18.

### E.4 RESULTS ON CIFAR-10-C, LEVELS 1-4

The following bar plots and tables are on levels 1-4 of CIFAR-10-C. The original distribution is the same for all levels, so are our results on the original distribution.

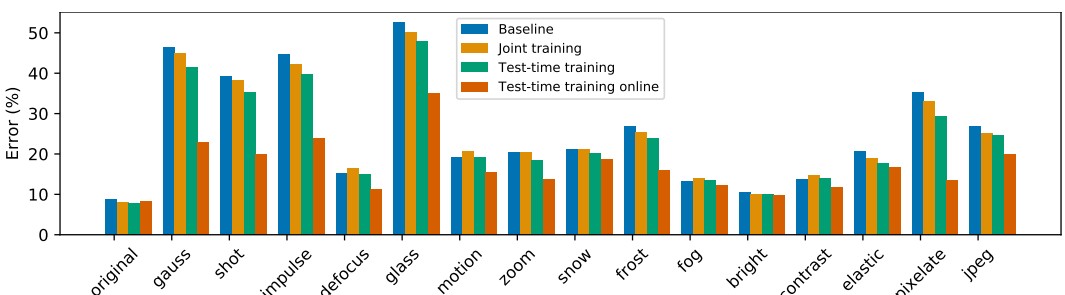

Figure 6: Test error (%) on CIFAR-10-C, level 4. See subsection 3.1 for details.

|      | orig | gauss | shot | impul | defoc | glass | motn | zoom | snow | frost | fog | brit | contr | elast | pixel | jpeg |
|------|------|-------|------|-------|-------|-------|------|------|------|-------|-----|------|-------|-------|-------|------|
| B    | 8.9  | 46.4  | 39.2 | 44.8  | 15.3  | 52.5  | 19.1 | 20.5 | 21.3 | 26.9  | 13.3 | 10.5 | 13.7  | 20.8  | 35.3  | 26.9 |
| JT   | 8.1  | 45.0  | 38.3 | 42.2  | 16.4  | 50.2  | 20.7 | 20.5 | 21.1 | 25.4  | 14.1 | 10.0 | 14.7  | 19.0  | 33.2  | 25.1 |
| TTT  | 7.9  | 41.5  | 35.4 | 39.8  | 15.0  | 47.8  | 19.1 | 18.4 | 20.1 | 24.0  | 13.5 | 10.0 | 14.1  | 17.7  | 29.4  | 24.5 |
| TTTO | 8.2  | 22.9  | 20.0 | 23.9  | 11.2  | 35.1  | 15.6 | 13.8 | 18.6 | 15.9  | 12.3 | 9.7  | 11.9  | 16.7  | 13.6  | 19.8 |
| ALP  | 16.5 | 21.3  | 20.5 | 24.5  | 20.7  | 25.9  | 23.7 | 21.4 | 24.2 | 23.9  | 42.2 | 17.5 | 53.7  | 22.1  | 19.1  | 18.5 |

Table 7: Test error (%) on CIFAR-10-C, level 4, ResNet-26.

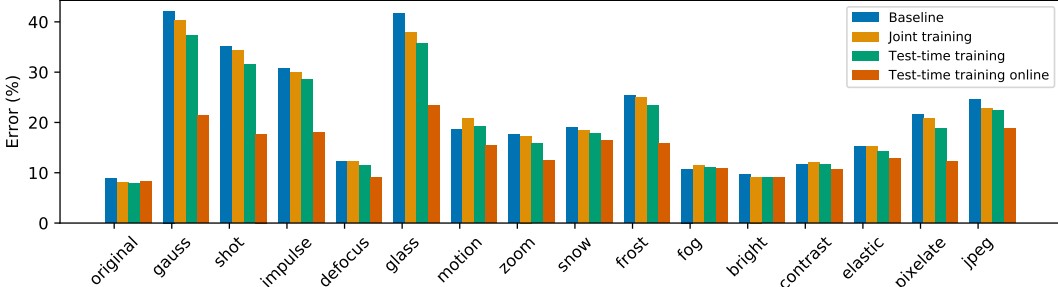

Figure 7: Test error (%) on CIFAR-10-C, level 3. See subsection 3.1 for details.

|      | orig | gauss | shot | impul | defoc | glass | motn | zoom | snow | frost | fog  | brit | contr | elast | pixel | jpeg |
|------|------|-------|------|-------|-------|-------|------|------|------|-------|------|------|-------|-------|-------|------|
| B    | 8.9  | 42.2  | 35.1 | 30.7  | 12.2  | 41.7  | 18.6 | 17.5 | 19.0 | 25.3  | 10.8 | 9.7  | 11.6  | 15.3  | 21.7  | 24.6 |
| JT   | 8.1  | 40.2  | 34.4 | 29.9  | 12.2  | 37.9  | 20.8 | 17.3 | 18.4 | 25.0  | 11.4 | 9.2  | 12.0  | 15.2  | 20.8  | 22.8 |
| TTT  | 7.9  | 37.2  | 31.6 | 28.6  | 11.5  | 35.8  | 19.1 | 15.8 | 17.8 | 23.3  | 11.0 | 9.1  | 11.6  | 14.3  | 18.9  | 22.3 |
| TTTO | 8.2  | 21.3  | 17.7 | 17.9  | 9.0   | 23.4  | 15.3 | 12.5 | 16.4 | 15.8  | 10.9 | 9.0  | 10.7  | 12.8  | 12.2  | 18.7 |
| ALP  | 16.5 | 20.0  | 19.3 | 20.5  | 19.2  | 21.2  | 24.0 | 20.5 | 20.9 | 24.2  | 30.1 | 16.6 | 39.6  | 20.9  | 17.8  | 18.0 |

Table 8: Test error (%) on CIFAR-10-C, level 3, ResNet-26.

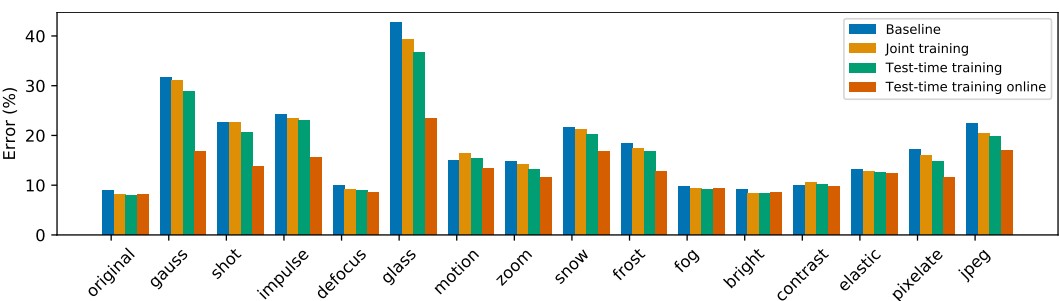

Figure 8: Test error (%) on CIFAR-10-C, level 2. See subsection 3.1 for details.

|      | orig | gauss | shot | impul | defoc | glass | motn | zoom | snow | frost | fog  | brit | contr | elast | pixel | jpeg |
|------|------|-------|------|-------|-------|-------|------|------|------|-------|------|------|-------|-------|-------|------|
| B    | 8.9  | 31.7  | 22.6 | 24.3  | 9.9   | 42.6  | 14.9 | 14.7 | 21.7 | 18.4  | 9.8  | 9.1  | 10.0  | 13.1  | 17.1  | 22.4 |
| JT   | 8.1  | 31.0  | 22.6 | 23.4  | 9.1   | 39.2  | 16.4 | 14.2 | 21.2 | 17.5  | 9.4  | 8.3  | 10.6  | 12.8  | 15.9  | 20.5 |
| TTT  | 7.9  | 28.8  | 20.7 | 23.0  | 9.0   | 36.6  | 15.4 | 13.1 | 20.2 | 16.9  | 9.2  | 8.3  | 10.2  | 12.5  | 14.8  | 19.7 |
| TTTO | 8.2  | 16.8  | 13.8 | 15.5  | 8.5   | 23.4  | 13.3 | 11.5 | 16.8 | 12.7  | 9.4  | 8.4  | 9.7   | 12.4  | 11.5  | 17.0 |
| ALP  | 16.5 | 18.0  | 17.2 | 19.0  | 17.8  | 20.7  | 21.2 | 19.3 | 19.0 | 20.1  | 22.4 | 16.3 | 29.2  | 20.3  | 17.4  | 17.8 |

Table 9: Test error (%) on CIFAR-10-C, level 2, ResNet-26.

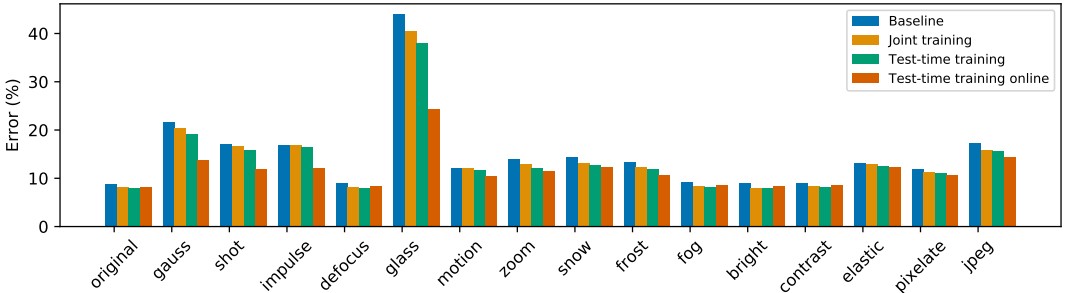

Figure 9: Test error (%) on CIFAR-10-C, level 1. See subsection 3.1 for details.

|      | orig | gauss | shot | impul | defoc | glass | motn | zoom | snow | frost | fog  | brit | contr | elast | pixel | jpeg |
|------|------|-------|------|-------|-------|-------|------|------|------|-------|------|------|-------|-------|-------|------|
| B    | 8.9  | 21.7  | 17.1 | 17.0  | 9.0   | 44.0  | 12.1 | 13.9 | 14.3 | 13.4  | 9.2  | 8.9  | 9.0   | 13.2  | 12.0  | 17.3 |
| JT   | 8.1  | 20.4  | 16.6 | 16.9  | 8.2   | 40.5  | 12.2 | 13.0 | 13.1 | 12.3  | 8.4  | 8.1  | 8.5   | 12.9  | 11.3  | 15.9 |
| TTT  | 7.9  | 19.1  | 15.8 | 16.5  | 8.0   | 37.9  | 11.7 | 12.2 | 12.8 | 11.9  | 8.2  | 8.0  | 8.3   | 12.6  | 11.1  | 15.5 |
| TTTO | 8.2  | 13.8  | 11.9 | 12.2  | 8.5   | 24.4  | 10.5 | 11.5 | 12.4 | 10.7  | 8.5  | 8.3  | 8.6   | 12.4  | 10.7  | 14.4 |
| ALP  | 17.0 | 16.8  | 17.6 | 16.8  | 20.9  | 18.7  | 19.0 | 17.3 | 17.5 | 17.4  | 16.1 | 18.4 | 20.4  | 17.0  | 17.2  |      |

Table 10: Test error (%) on CIFAR-10-C, level 1, ResNet-26.

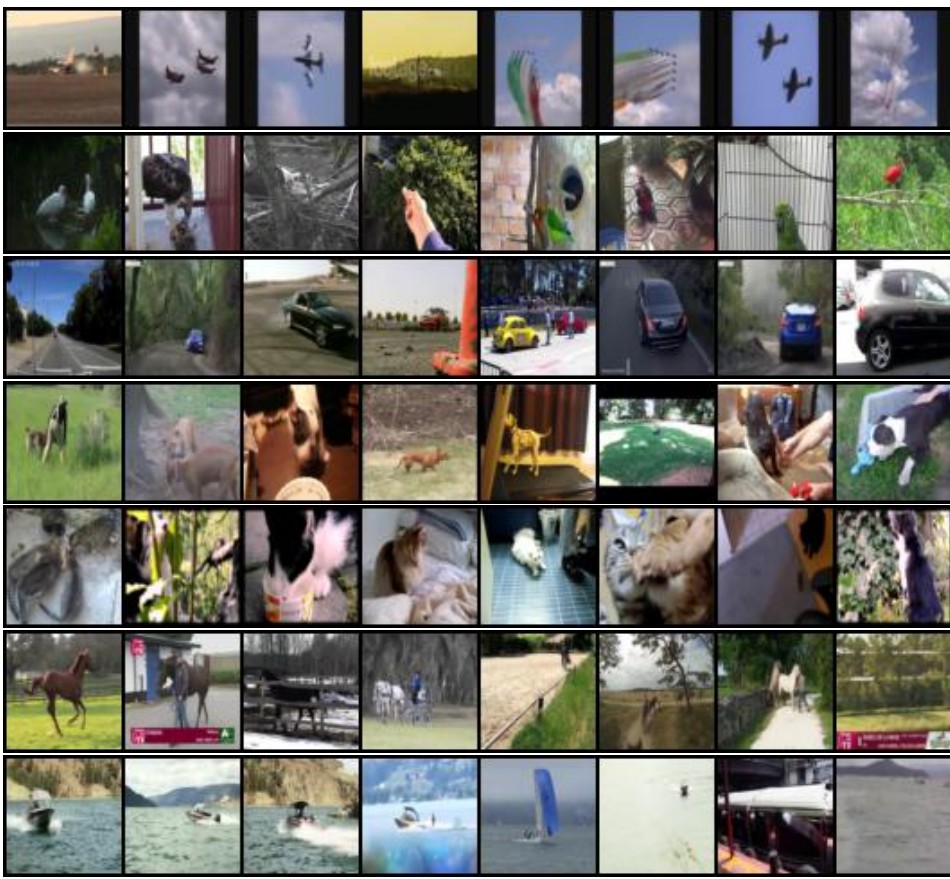

Figure 10: Sample Images from the VID dataset in subsection 3.2 adapted to CIFAR-10. Each row shows eight sample images from one class. The seven classes shown are, in order: airplane, bird, car, dog, cat, horse, ship.

