# OpenReview forum: "Test-Time Training for Out-of-Distribution Generalization"
_ICLR.cc/2020/Conference — Reject_

### Official Review · AnonReviewer1 · 2019-10-19
**Official Blind Review #1**

**Rating:** 6

**Review:**

The authors propose a method for adapting model parameters by doing self-supervised training on each individual test example. They show striking improvements in out-of-domain performance across a variety of image classification tasks while preserving in-domain performance; the latter is a marked difference from other robustness procedures which tend to sacrifice in-domain performance for out-of-domain (or adversarial) performance. These results are exciting, and I believe that this proposed test-time training method will spur a significant amount of further research into similar approaches.

The paper is well-written and the experiments are thorough, so I have no major concerns. Some remaining questions about the proposed approach are:

1) How sensitive is test-time training to hyperparameters like splitting the parameters at the right location (i.e., the particular partitioning of $\theta$ into $\theta_e$, $\theta_s$, and $\theta_m$), or to the learning rate? Is there a good way to pick these hyperparameters, given that evaluation is on an out-of-domain distribution that we assume we do not have access to at training time? The paper proposes a particular split of parameters and a particular learning rate and number of steps (which differs for standard vs. online training). How were those chosen?

2) How does test-time training compare to methods that assume access to the test distribution? I understand that a big benefit is that test-time training does not need to see the entire distribution (unlike standard domain adaptation approaches). But in cases where we do get to see parts of the test distribution -- say some unlabeled examples from it, or even some labeled examples -- how does test-time training compare? For example, should we see test-time training as providing the benefits of domain adaptation even when we're unable to access the unlabeled test distribution; or should we see it as doing something beyond what standard domain adaptation methods do, even when we have access to the unlabeled test distribution?

Minor comments, no need to respond:
a) There are several minor typos in the paper, e.g.: p1, "prediciton"; eqn 8, v; p8, "address"; p9, "orcale"; appendix A, missing ref.
b) The discussion in Appendix A seems a bit speculative and opinionated. For example, it is not obvious that one has to fall back on the space of all possible models in order to represent test-time training with a single gradient step as a fixed model. The discussion is useful but in my subjective opinion could be toned down; the experiments and discussion in the main paper are strong and less speculative.

===

Edit: Thank you for the response. The discussion about hyper-parameters makes sense. My rating edit comes from the realization that the performance improvements obtained are almost entirely from the "online" version, which gets to see the test distribution. So I think the baselines are in lacking in that sense: as a straw man baseline for example, one could simply run the normal model on half of the test set, and then use those observed test examples to do some other sort of domain adaptation training.

**Experience Assessment:**

I have published one or two papers in this area.

**Review Assessment: Checking Correctness Of Derivations And Theory:**

I assessed the sensibility of the derivations and theory.

**Review Assessment: Checking Correctness Of Experiments:**

I assessed the sensibility of the experiments.

**Review Assessment: Thoroughness In Paper Reading:**

I read the paper thoroughly.

---

> ### Author Response · Authors · 2019-11-14
> **Thank you and answers to your questions**
>
> Thank you for your thoughtful comments. Here we answer your questions:
>
> 1. We do not need information about the new distribution to choose the hyper-parameters. There are three hyper-parameters for test-time training: the splitting point, the learning rate and the number of steps. We select our splitting point to maximize performance of the joint-training baseline on the original distribution. The learning rate for test-time training is the same as during the last epoch of regular training; intuitively, this lets the model keep learning at the rate it has been accustomed to. Both these hyper-parameters can be selected without any knowledge of the test distribution, as done in our experiments. Empirically, we observe that performance is rather insensitive to the number of test-time training iterations once the self-supervised loss (on the test instance) converges. This also makes intuitive sense because once convergence is reached, the gradient from the self-supervised loss is small anyways. For the standard version, 10 steps is more than enough to reach convergence; we have in fact experimented with taking more steps and observe no difference in performance beyond random variations. For the online version, 1 step is in fact enough to reach convergence because the algorithm has already seen many previous samples from the same distribution. Practitioners can easily observe convergence of the self-supervised loss with the information revealed during testing.
>
> 2. Thank you for acknowledging that we do not need to see the new distribution. In the case where some unlabeled samples are indeed available (as in the setting of unsupervised domain adaptation), yes they should be taken advantage of through methods for that setting. Test-time training can be applied on top of the model that has used these samples.
>
> You minor comments: we really appreciate your careful reading and have incorporated them in our revision.

---

### Official Review · AnonReviewer2 · 2019-10-20
**Official Blind Review #2**

**Rating:** 6

**Review:**

The paper proposes test time training, a method that uses an auxiliary task to provide a kind of loose supervision during test time. I say loose supervision because the theory suggests this is only useful when the gradients of the main and auxiliary losses are positively correlated.

I'm not sure I totally believe that this is a method of out-of-distribution generalization, but rather it helps adjust for corruptions and modest dataset shifts which is an important problem itself. I suspect test-time training is fundamentally better suited for the latter because in general the assumption of correlation between the loss gradients cannot hold in the test if we allow for large shifts (like the airplane class in the video experiment). I note that the authors are aware of this limitation (page 5, last paragraph).

My main problem with this paper is that the more fine-tuned labels get (like the density of a tumor), the harder it gets to create auxiliary tasks. This will be a significant problem when the samples at test-time only share the highest level common characteristics with the true dataset; (like rotations do not impact or density of tumor, like face detection and similar fine-feature-based tasks).

That said, I do appreciate the experimental results which show promise; especially the CIFAR-10.1 results. So I'm inclined toward accepting this paper. I would be more so inclined if the authors could provide a reasonable categorization of tasks where this method is expected to be applicable. I ask this because, for practitioners, it is near impossible to verify the positive correlation of gradients assumption. If there are high-level targets that the distribution and/or task must satisfy that can act as indicators for applying this method, I believe that would be a valuable addition to the paper.


**Experience Assessment:**

I have read many papers in this area.

**Review Assessment: Checking Correctness Of Derivations And Theory:**

I assessed the sensibility of the derivations and theory.

**Review Assessment: Checking Correctness Of Experiments:**

I assessed the sensibility of the experiments.

**Review Assessment: Thoroughness In Paper Reading:**

I read the paper at least twice and used my best judgement in assessing the paper.

---

> ### Author Response · Authors · 2019-11-14
> **Thank you and answers to your questions**
>
> Thank you for your positive feedback. We agree our method “helps adjust for corruptions and modest dataset shifts”. We understand the term out-of-distribution to broadly include distribution shifts of all kinds, including the small and modest ones in our experiments. To avoid this confusion, however, we now use the term distribution shifts instead of out-of-distribution in the revision.
>
> You are  worried that “the more fine-tuned labels get (like the density of a tumor), the harder it gets to create auxiliary tasks.” Even if the main task is highly specialized (e.g. the tumor density example), the auxiliary task can be fairly general (e.g. rotation). The self-supervised task only needs to share *features* with the main task without actually solving it, and features in computer vision can be as general as edges and shades. In ImageNet for example, improvements are aggregated across very specific problems such as distinguishing between 280 kinds of birds and 62 kinds of lizards, but rotation suffices for self-supervision.
> In addition, the theoretically sufficient condition of our method -- gradient correlation -- is agnostic to the size of the label space; even if the label space is large (ImageNet with 1000 classes) or infinite (regression), using rotation still performs well.
>
> You also asked for “...a reasonable categorization of tasks where this method is expected to be applicable.” We can provide reasonable rules of thumb for both the standard and the online version of our method. Standard: The self-supervised task, e.g. rotation, is both well defined and non-trivial on the new domain (in the sense discussed in Section 3.2). Online: All the test samples in the sequence are from the same (new) test distribution. Both conditions are easy to check in practice. Empirically, our method was shown to be effective in all of the experiments where these rules are met.

---

### Official Review · AnonReviewer3 · 2019-10-26
**Official Blind Review #3**

**Rating:** 6

**Review:**

The motivation is to increase accuracy of CNNs with unseen (unknown) distribution shifts. To this end, instead of the recent approach of training-time self-supervision, they adopted it for test-time [limited novelty]. More precisely, for a classification task, they considered two headed neural network with one head for main classification task and another for an auxiliary task (e.g. predicting rotation degree of rotated images). The feature extractor (up to k layers) is shared between these tasks thus updated using the two tasks, while two heads are updated according to each task. In test-time, the samples (drawn from shifted data distributions) are being used for updating the shared feature extractor through the auxiliary task. They have investigated their test-time approach in a series of experiments in online and offline settings on synthetic shift in data distribution of image classification tasks. The distributional shifts are synthesized adding some non-adversarial perturbation to clean images. The authors proof that their test-time approach can lead to lower error rate on given test-time samples when the underlying learning model is a linear regression model.
The authors incorrectly interchange OOD with domain shifts in their manuscript. The common usage of OOD set is to capture novel samples that are not properly following the training set distribution, e.g. from different concepts than the set of classes given in the training set. With an OOD set, a robust model dealing with it should be able to detect whether an instance is in-distribution or OOD, making a special decision for the latter case (e.g., rejecting the instance).
In domain shift, we look at the scenario where test objects are the same as used for training the model (i.e., correspond to one of the classes the model is processing), but might be perturbed or coming having a different distribution (e.g. SVHN for MNIST or vice versa). The approaches for domain shift concern to improve robustness of CNNs to such shifts in data distribution. Accordingly, the title of the paper inaccurately reflects of the claim of the paper and is misleading, this paper is not on learning with out-of-distribution instances.
The other important point is about catastrophic forgetting phenomena in online setting of their approach, which was not addressed thoroughly in the paper. How not to forget what the model has previously learnt a test-time training? I see this somewhat has been empirically shown in Fig 2 with accuracy on the original data, but what is the mechanism not to forget what have being learned so far?
Besides generalization enhancement, the advantage of test-time self-supervision over training-time joint self-supervised is not clear for the readers, particularly considering the problem of the pitfall of catastrophic forgetting phenomena in test-time training. This pitfall does not exist for training-time joint self-supervised approach. What are the advantages of this approach?
The claims (about synthetic shift in distribution shift) are well supported in a series of experiments, where the distribution shifts are synthesized using adding some non-adversarial perturbation to clean images. However, the other common experiments on real shifts in distribution (e.g. SVHN for MNIST and vice versa, and those performed in Volpi et al  2018) are missing, which can help the paper to being better supported and justified for its practicality.
[Volpi et al  2018]: Generalizing to Unseen Domains via Adversarial Data Augmentation, NIPS 2018
I found the paper rather difficult to follow and not very coherent in its organization. The main idea is fundamentally simple, but it is still difficult to get it from the text. It needed me 2-3 readings before really getting the point of the paper.
** Update ** I read other reviews and comments. The answer of the authors to my comments are somehow satisfactory, especially the point of changing from "out-of-distribution" to "domain shift", which avoid some confusion. I upgraded my rating to a "weak accept".

**Experience Assessment:**

I have published one or two papers in this area.

**Review Assessment: Checking Correctness Of Derivations And Theory:**

I assessed the sensibility of the derivations and theory.

**Review Assessment: Checking Correctness Of Experiments:**

I assessed the sensibility of the experiments.

**Review Assessment: Thoroughness In Paper Reading:**

I read the paper at least twice and used my best judgement in assessing the paper.

---

> ### Author Response · Authors · 2019-11-14
> **Thank you and answers to your questions**
>
> Thank you for your feedback. It appears our use of the term “out-of-distribution” caused some confusion. Our algorithm works on what you call “domain shifts” and does not deal with out-of-distribution detection. This terminology difference is secondary to the main contribution of the paper. In the revision, we use the term distribution shifts instead of out-of-distribution.
>
> Forgetting has minimal impact on the performance of our method. This has been shown empirically in the paper, as you recognize in your review. Our model is jointly trained on both tasks, so our corrections during test-time training are tiny. This is in contrast to continual learning, where forgetting arises because the tasks have never been jointly trained and are learned one-by-one from scratch.
>
> We also conducted the following experiment. The widely adopted oracle in continual learning is to jointly train on all the tasks. For test-time training, we experiment with the analogous oracle by mixing in training of the main task (on the training set) with the self-supervised task (on the test instance). This modification of our method takes a very long time to run, but should exhibit as little forgetting as possible. On all the benchmarks in CIFAR-10-C level 5, the standard and online versions of our method have the same performance both with and without this modification (up to random fluctuation), demonstrating that the impact of forgetting is minimal.
>
> Regarding experiments following [Volpi et al 2018]: Digit datasets (e.g. MNIST), especially those of small image dimensions, are generally not good fits for self-supervised learning. Rotation in particular can be poorly defined for digits. We experiment with real distribution shifts of natural scenes in the paper.

---

### Public Comment · ~Jiao_YU_Shen1 · 2019-10-19
**Some questions**

First of all, I'd like to thank the author for the work, which inspire me to the generalization capability of real-world perturbation. I have few questions regarding the work.

1. Based on my understanding, the formulation is very similar to the jigsaw puzzle published in CVPR2019 which also tackled the out of distribution generalization task. My impression is that this work adopted a image rotation for auxiliary task, while jigsaw puzzle adopted image patch shuffling as auxiliary task.  May I know whether there are any other differences in high level?

2. When talking about out of distribution generalization, may I know whether the proposed problem can tackle the case where the augmentation types are unknown in advance (especially for testing samples). I guess this might not be a problem of jigsaw puzzle as it did not require testing data for parameter updating.

3. I have difficulty understanding the theory 1 part. Based on the coarse of deep learning, the network is usually not convex. May I know whether there are any motivation by assuming x, y, l are all convex in \theta?

4. My research focus currently is on distribution based for generalization problem. May I know how this problem differentiate with the MMD based domain generalization method? such as the paper "Exploiting Low-rank Structure from Latent Domains for Domain Generalization" published in ECCV14 and other related works.

Finally, I'd like to thank the author again for this inspiring work. I am looking forward to the explanation which I think will definitely help me with my future research.

---

> ### Author Response · Authors · 2019-10-19
> **Answers**
>
> Thank you for your kind words and we are glad to be able to inspire your research. Here we answer your questions:
> 1. We cite and discuss this CVPR'19 paper in the related work section. Their method roughly corresponds to our joint training baseline, which we always compare with empirically. At a high level, the difference is simply that they do not perform test-time training i.e. modify the model parameters at test-time, which is exactly what we claim to contribute.
> 2. We assume you meant to say "corruption types" instead of "augmentation types", and "testing labels" instead of "testing data". Then the case you described, of corruption types unknown in advance, is exactly the case we are trying to solve.
> 3. The motivation is to make the analysis tractable, since we currently do not have sufficient tools to theoretically reason about realistic networks.
> 4. We are not sure we understand your question, which asks us to compare the "generalization problem" with MMD-based methods; the comparison between a problem and a class of methods seems undefined. If you are looking for a comparison between the "generalization problem" and the problem of unsupervised domain adaptation, which often uses MMD-based methods and knows the target distribution in advance in the form of many unlabeled samples, we discuss this in the related work section. If you are looking for a comparison between our method and MMD-based methods, the most immediate difference is that the mean discrepancy is degenerate for a single sample thus cannot be used for test-time training.

---

> > ### Public Comment · ~Jiao_YU_Shen1 · 2019-10-20
> > **Some questions**
> >
> > Thanks very much. I will study the paper again based on your explanation to me.

---

### Decision · Program_Chairs · 2019-12-19

**Decision:**

Reject

**Comment:**

The paper is on a new approach approach to transductive learning. Reviewers were a bit on the fence. Their most important objection is that the performance improvements that the authors report almost entirely come from the "online" version, which basically gets to see the test distribution.  That contribution is nevertheless, in itself, potentially interesting, but I was surprised not to see comparison with simple transductive learning from semi-supervised learning, learning with cache, or domain adaptation, e.g., using knowledge of the target distribution to reweigh the training sample, or [0], on using an adversary to select a distribution consistent with sample statistics. I encourage the authors to add more baselines, analyze differences with existing approaches, and, if their approach is superior to existing approaches, resubmit elsewhere.

[0] http://papers.nips.cc/paper/5458-robust-classification-under-sample-selection-bias.pdf